# Semen Thresholds of Normality Established by the WHO Do Not Reveal Genome Instability—A Potential Occult Male Factor

**DOI:** 10.3390/genes14020239

**Published:** 2023-01-17

**Authors:** Usha Punjabi, Ilse Goovaerts, Kris Peeters, Diane De Neubourg

**Affiliations:** 1Centre for Reproductive Medicine, Antwerp University Hospital, Drie Eikenstraat 655, 2650 Edegem, Belgium; 2Department of Reproductive Medicine, Antwerp Surgical Training, Anatomy and Research Centre (ASTARC), Faculty of Medicine and Health Sciences, University of Antwerp, Drie Eikenstraat 655, 2650 Edegem, Belgium

**Keywords:** sperm DNA fragmentation, semen parameters, chromatin maturity, sperm aneuploidy, genome instability, male factor, normozoospermia

## Abstract

Semen parameters are unable to inform on the function or fertilizing capacity of the male gamete. Standardized methods are provided by the WHO but, the lower reference limits have reduced sensitivity to predict chances of conception. Subfertile men may be falsely classified as “normal” and a male factor contributing to genome instability may be overlooked. Semen parameters, sperm DNA fragmentation (SDF), sperm chromatin maturity and stability, and sperm aneuploidy were assessed in fertile (F), subfertile normozoospermic (SN) and subfertile non-normozoospermic males (SN-N). Standardized assays employing flow cytometry were used to detect genome instability. Sperm DNA fragmentation did not differ significantly whether the semen samples were from a fertile (F), subfertile normozoospermic (SN) or subfertile non-normozoospermic male (SN-N). Chromatin decondensation was significantly reduced and hyperstability significantly increased in the SN group as compared to the F group. The frequency of diploidy was significantly different in the three study groups with significance between F and SN and between F and SN-N groups. Subfertile men with normal semen parameters are often excluded from extensive genetic testing. Genome instability might be an independent attribute of semen quality detecting problems not seen with semen analysis alone.

## 1. Introduction

Infertility affects approximately 15% of couples worldwide, with studies revealing a wide range of estimated contributions of male infertility (5–35%) [1]. Conventional semen analysis is acknowledged as the cornerstone of the assessment of male infertility. However, unfortunately, semen parameters are only partly able to inform on the function or fertilizing capacity of the male gamete or predict chances of conception, both in vivo [2] and in vitro [3]. This is especially important in cases with unexplained infertility, where men with normal sperm parameters still experience reproductive failure due to poor/absent fertilization, poor embryo development, implantation failure, or pregnancy loss [4,5]. Although WHO has provided well-defined, standardized methods, unfortunately, the fifth edition of WHO guidelines use a one-tailed approach, basing the reference levels on the fifth centiles [6,7]. These values are often wrongly quoted indicating ‘fertility’ if a sample exceeds all lower limits and ‘infertile’ if it falls below the reference for one or more parameters [7]. The recent sixth edition [5] substantially added new data to the distribution values of the fertile male [8] but, the reference values have remained unchanged.

A temporal trend in semen quality has been observed giving evidence for decreasing quality of semen during the past 50 years [9]. Recently, it was indicated that impaired semen quality is associated with shorter life expectancy and increased long-term morbidity [10,11,12,13], emphasizing the significance of diagnosing male infertility. Consequently, there is a need for adjusted genetic screenings and improved semen biomarkers to assess the likelihood of a healthy offspring. There is evidence to support the notion that male fertility is not limited to defective spermatogenesis, but rather that spermatogenetic defects may be just one manifestation of a more systemic problem [10,14,15]. The occurrence of genome instability associated with male infertility is receiving more and more attention. DNA damage, including single- and double-strand breaks, is the most important factor that induces genome instability. Defective spermatogenesis, abnormalities in chromatin remodelling and abortive apoptosis are the major factors affecting the integrity of testicular sperm DNA, while testicular and post-testicular oxidative stress might also induce DNA damage [16,17].

During spermiogenesis, the transition of round spermatids into mature spermatozoa involves the replacement of nucleosome histones by protamines [18]. Chromatin packaging also requires endogenous nuclease activity to loosen the chromatin by histone hyper-acetylation and introduction of breaks by topoisomerase II, capable of both creating and ligating breaks. These combined DNA-condensing activities may optimize the strand repair process, emphasizing the link between altered sperm DNA condensation and DNA fragmentation [19]. Abnormally high amounts of histones in sperm are associated with decreased fertility and increased risk of embryonic failure after fertilization [20]. Therefore, histone retention and protamine deficiency in sperm are hallmarks of certain forms of idiopathic infertility [21,22,23,24].

Conversely, during fertilization the protamines are removed, the sperm nucleus condenses and the DNA combines with egg histones, forming the male pronucleus. Defects of sperm chromatin which prevent or delay chromatin decondensation can be expected to prevent normal development of the male pronucleus. The oocyte has an important but limited DNA repair capacity, largely efficient in relation to DNA strand breaks [25]. The oocyte capacity to repair sperm maturity or stability defects is rather limited. Therefore, determination and correction (when possible) of decondensation are of paramount importance in assisted reproductive technologies [26].

However, many men have no known cause of their infertility (idiopathic). Chromosomal aberrations, either numerical or structural, can have profound effects on fertility [27,28]. Infertile males produce gametes with a higher rate of chromosomal abnormalities than those found in the general population [29]. Chromosome stability is of crucial significance in cell division and propagation. An abnormal number of chromosome(s) during unbalanced cell separation at cell division is associated with almost all solid tumour cancers.

Male infertility is a multifactorial disease that can be caused by a wide variety of inherited (genetic) and acquired (lifestyle) factors. While fertility is defined as the capacity to establish a clinical pregnancy [30,31], ‘normozoospermia’ refers to the contemporary presence of sperm concentration, motility and morphology above the fifth centile reported by WHO (2021). However, semen analysis and the new WHO manual widen the perspective of semen examination not only for medically assisted reproduction but also as a tool to understand functions and disorders of male reproductive organs and general sexual and reproductive health [32].

We were interested in identifying a subtle male factor, especially in individuals who met a normal semen profile which could potentially influence the reproductive potential of the individual. With semen parameters well above the WHO lower reference intervals, would there be detectable sperm genome instability in normozoospermics?

## 2. Materials and Methods

### 2.1. Study Protocol

The study was monocentric, cross-sectional, prospective and partly retrospective. Recruitment and data collection occurred during different phases. Accordingly, several projects were approved by the Ethical Commission of the Antwerp University Hospital and the University of Antwerp.

Sperm DNA fragmentation in a fertile population conducted between October 2017–October 2020, approved on 26 June 2017, ref. no: 17/24/285 (Belgian registration no: B300201732872).

Sperm DNA fragmentation in an infertile population was conducted between October 2017–October 2020 and approved on 11 August 2017 (Belgian registration no: B300201733352).

Chromatin maturity and stability were conducted between January 2020–March 2020 and approved on 13/01/2020, ref. no: 19/51/629.

Sperm aneuploidy retrospective data collection was conducted between January 2014–December 2016 and approved on 6 July 2020, ref. no: 20/26/350.

### 2.2. Participants

The study population comprised a cohort of patients (18–40 years old) undergoing their first infertility diagnosis and treatment at the Centre for Reproductive Medicine, Antwerp University Hospital, Belgium. Subjects were excluded on the following terms: azoospermia (no spermatozoa) and cryptozoospermia (few hidden spermatozoa). Based on the lower reference limits established by the WHO [6], the study group was further split into:

Subfertile group normozoospermic (SN): sperm concentration ≥ 15 million/mL; progressive sperm motility ≥ 32% and sperm morphology ≥ 4%.

Subfertile group non-normozoospermic (SN-N): sperm concentration < 15 million/mL and/or <32% progressive motility and/or < 4% morphology.

Another group of fertile men (who achieved pregnancy within 12 months of unprotected coitus) and sperm donors (who had self-fathered children or had achieved pregnancies within the donor program of the clinic) were included as a control group (F). All subjects had given written informed consent for participation.

### 2.3. Semen Analysis

Semen samples were collected at the laboratory, and the analysis initiated within 60 min after ejaculation conforms to international standards of ISO 15189 (International Standards Organization, 2012). Standard semen parameters including sperm concentration, motility and morphology were determined using WHO [6] recommendations, and complying with the checklist for acceptability reported by Björndahl et al. [33]. All staff members were trained in basic semen analysis (ESHRE—European Society for Human Reproduction and Embryology Basic Semen Analysis Courses) [34,35] and participated regularly in internal and external quality control programs (Institute of Public Health, Belgium and ESHRE External Quality Control Schemes, Sweden) [36]. Only complete samples were included.

### 2.4. Sperm DNA Fragmentation (SDF)

Assessment of SDF was performed using the terminal deoxynucleotidyl transferase-mediated deoxyuridine triphosphate nick-end labelling (TUNEL assay) described by Mitchell et al. [37]. Briefly, spermatozoa were incubated for 30 min at 37 °C with LIVE/DEAD^®^ Fixable Dead Cell Stain (far red) (Molecular Probes, Life technologies, Eugene, OR, USA) after which the cells were washed 2x with phosphate-buffered saline (PBS, GIBCO Life technologies, Paisley, UK) before being incubated with 2mM dithiothreitol (DTT, Sigma-Aldrich, Overijse, Belgium) for 45 min. Following this, the samples were washed 2x in PBS and fixed in 3.7% formaldehyde (Sigma-Aldrich, Belgium) for 20 min at 4 °C. As storage of the sample at 4 °C affects reproducibility [38] the assay was carried out directly on fresh semen samples without storage. For the assay, the spermatozoa were washed 2x and centrifuged before being resuspended in 500 µL of fresh permeabilization solution (100 mg Sodium citrate, 100 µL Triton X–100 in 100 mL dH_2_O) and incubated for 5 min at 4 °C. The cells were washed 2x with PBS. The positive control samples were treated with 5 µL of DNase I (Qiagen, Germany) 1500 Kunitz Units for 30 min at room temperature. The assay was performed using the fluorescein In Situ Cell Death Detection Kit (Roche Diagnostics, Mannheim, Germany) using Accuri C6 flow cytometer (BD Sciences, Erembodegem, Belgium). For each sample, 5000–10,000 events were recorded at a flow rate of 35 µL/min.

DNA fragmentation was analysed in the total sperm sample (total SDF) comprising viable and nonviable sperms as well as in the vital fraction (vital SDF) thereby analysing only viable sperm. The method was standardized and cut-off values were defined [39,40].

### 2.5. Sperm Nuclear Chromatin Condensation and Decondensation Assessment

Sperm chromatin condensation and decondensation were evaluated according to the procedure by Molina et al. [41]. In brief, an aliquot of the semen sample is added to two test tubes containing 1ml Tris buffer to achieve a concentration of ±5 × 10^6^ spermatozoa per ml. After washing and centrifugation, one test tube was treated with the DNA-intercalating dye propidium iodide (Sigma-Aldrich, Belgium, PI, 50 µg/mL) followed by flow cytometric evaluation of the PI fluorescence intensity on a cell-per-cell basis. This was carried out on a Facscan (BD Biosciences, Erembodegem, Belgium) equipped with standard excitation and emission optics. The resulting PI fluorescence frequency distribution reflects the status of DNA condensation in the measured nuclei. The second test tube was treated with 1% sodium dodecyl sulphate (SDS, Sigma-Aldrich, Belgium) plus 6 mmol/L ethylene diamine tetra acetic acid (EDTA, Sigma-Aldrich, Belgium) decondensing solution in borate buffer (Sigma-Aldrich, Belgium) for 5 min. Following this, the sample was washed and centrifuged before using PI. Approximately 3000–9000 cells for each sample were analysed. The mean channel of fluorescence was used to analyse the accessibility and consequently, the degree of staining of sperm DNA with PI and the following flow cytometry parameters were analysed:Condensed chromatin—histones replaced by protamines transforming the nucleus into a highly compact structure.Hypocondensed chromatin—insufficient chromatin condensation or a potential condition of underprotamination rendering the paternal genome susceptible to damage.Decondensed chromatin—ability of compacted chromatin to decondense in vitro after SDS and EDTA treatment.Hypercondensed chromatin—resistance to decondensation achieving a state of hyperstability making the paternal genome unavailable for further fertilization.

The method was standardized and cut-off values were defined for all chromatin parameters [42].

### 2.6. Fluorescence In Situ Hybridization (FISH) Analysis

Sperm samples were washed with phosphate-buffered saline (PBS; Gibco; Life Technologies, Paisley, UK) and the resulting pellet was fixed in Carnoy’s solution (methanol/acetic acid, 3:1; Merck, Belgium). The fixed specimens were stored at −20 °C until further processing. Cytogenetic analysis of 5 chromosomes: chromosome 13, 18, 21, X/Y was performed according to Vegetti et al. [43]. Briefly, the fixed spermatozoa were spread on a slide and air-dried. For nuclear decondensation, the air-dried slides were washed in 2x saline citrate solution (20X SSC, Invitrogen) and incubated in 1 mol/L Tris buffer containing 25 mmol/L DTT (Sigma-Aldrich, Belgium). Following decondensation, the slides were washed in 2x SSC and dehydrated through an ethanol series and air-dried. A two-colour FISH using locus-specific probes for chromosomes 13 (spectrum green) and 21 (spectrum red) and a three-colour FISH with centromeric probes for chromosomes X (spectrum green), Y (spectrum red) and 18 (spectrum blue) was performed. Probes were supplied by Vysis (Abbott Laboratories) and the FISH protocol was performed according to Vysis. Slides were observed using an Axioplan epifluorescence microscope (Leica, Wetzlar, Germany) with appropriate filter sets. For each probe, a maximum of 1000 spermatozoa were counted per patient. Only intact spermatozoa with clear hybridization signals were scored, disrupted or overlapping spermatozoa were excluded. Sperm nuclei were scored nullisomic when no signal for the investigated chromosomes was seen. Sperm nuclei were considered disomic when two similar signals of the same colour were observed. Finally, sperm nuclei were considered diploid when two signals for each tested chromosome were exhibited in intact spermatozoa. WHO [5] values for sperm disomy and our own fertile population levels for nullisomy were adapted.

Genome instability in sperm can be accessed via SDF, chromatin maturity and aneuploidy which might potentially lead to a male factor infertility as schematically presented in Figure 1.

### 2.7. Statistical Analysis

Statistical analyses were conducted using Medcalc^®^ version 20.027 (MedCalc Software Bv, Oostende, Belgium). Descriptive statistics (mean, standard deviation (SD) and range) are reported for the patient characteristics, semen parameters, SDF parameters, chromatin parameters and chromosome aneuploidy. Spearman correlation was calculated between SDF parameters, chromatin maturity and stability, sperm aneuploidy and semen parameters.

Semen variables were necessarily back-transformed after logarithmic transformation. Data distributions were evaluated by the Kolmogorov–Smirnov test. To assess differences in continuous variables between 2 groups, the Unpaired Student T-test was used in case of normal distribution and the Mann–Whitney test was used in case the data were not normally distributed. Differences in continuous variables between 3 or more groups were assessed using the ANOVA and Kruskal–Wallis tests. If significant, the groups were compared pairwise using a post-hoc test. Comparisons of data distributions between the fertile and subfertile groups were conducted by constructing receiver operating characteristic (ROC) curve analysis. When considering the frequency of aneuploidies, a 99.9% upper limit of normality was calculated using the non-parametric percentile method for the control fertile population. Fischer’s exact test was applied for frequencies of aneuploidies between the SN and SN-N study groups. For all statistical tests, differences with a *p* value < 0.05 were considered significant.

## 3. Results

Semen parameters were assessed in 753 samples (fertile and subfertile). In 458 samples (60.8%) semen parameters were normal while one or more abnormalities were noted in the rest. In an additional 121 samples, chromatin maturity was determined and sperm aneuploidy in another 195 samples with a normal somatic karyotype, together with semen parameters.

Using ROC curve analysis (sensitivity 95.5%, specificity 21.6%, *p* = 0.037) threshold criteria for age were determined using the Youden J index (≤40 years). The mean male age included was 31.4 ± 5.4 (range: 18–40) years with no significant difference (*p* = 0.092) between the fertile and subfertile groups.

### 3.1. Semen Parameters in Fertile, Subfertile Normozoospermic and Subfertile Non-Normozoospermic Males

In the semen samples analysed (F = 42; SN = 416; SN-N = 295), there was a significant difference (*p* < 0.001) between the three study groups as far as sperm concentration, motility and morphology was concerned. With the significance being more pronounced between the F and SN-N and between SN and SN-N groups for sperm concentration and progressive motility. Sperm morphology in the SN group, although within the normal threshold values, was significantly lower than the F group (Figure 2).

### 3.2. SDF Parameters in Fertile, Subfertile Normozoospermic and Subfertile Non-normozoospermic Males

Total SDF was significantly negatively correlated (r = −0.14; *p* ≤ 0.001) with progressive motility and vital SDF (r = −0.08; *p* ≤ 0.001) with sperm concentration (Figure 3).

Total SDF did not differ significantly (*p* = 0.416) whether the semen samples were from an F (8.1 ± 5.8; *n* = 42), SN (11.7 ± 7.8; *n* = 416) or SN-N (12.2 ± 10.2; *n* = 295) study group. The same was also the case (*p* = 0.688) in the vital SDF fractions (F = 1.3 ± 1.4; SN = 1.4 ± 1.6 and SN-N = 2.0 ± 3.0, respectively). Figure 4 reveals the distribution of SDF parameters in the three study groups.

### 3.3. Chromatin Maturity and Stability in Fertile, Subfertile Normozoospermic and Subfertile Non-Normozoospermic Males

There was a significant difference (*p* ≤ 0.001) between the three study groups as far as chromatin condensation, chromatin decondensation in vitro, hypercondensation and hypocondensation was concerned.

The significance was more pronounced between the F and SN-N and between SN and SN-N groups for all four chromatin parameters (Figure 5). Chromatin decondensation was significantly reduced and hypercondensation significantly increased in the SN group as compared to the F group (Figure 5).

### 3.4. Sperm Aneuploidy in Fertile, Subfertile Normozoospermic and Subfertile Non-Normozoospermic Males

The frequency of sperm aneuploidy in males with a normal karyotype concerning chromosomes 13, 18, 21 and X/Y were not significantly different in the three study groups (Table 1).

The frequency of diploidy, on the other hand, was significantly different in the three study groups (*p* = 0.011). With the significance being between F and SN and between F and SN-N groups (Figure 6).

The frequencies of sperm nullisomy, disomy and diploidy were considered when alteration was above the upper limit of normality of the F group. In short, for chromosome 13, the upper values for nullisomy and disomy were 0.6% and 0.4%; for chromosome 18, 0.3% and 0.7%; and for chromosome 21, 0.4% and 0.2%, respectively. For sex chromosome nullisomy the upper value was 1.2% and the disomy was 0.8% (XX), 0.4% (XY) and 0.3% (YY). Autosomal aneuploidy and diploidy had an upper limit of 1.3% and sex chromosome aneuploidy was 1.6%. Figure 7 shows the frequency and percentages of aneuploidy in the SN and SN-N study groups. A total of 47.3% (96/203) had aneuploidy in the subfertile groups; 49 in the SN (41.9%) and 47 (54.7%) in the SN-N (*p* = 0.0718) groups, respectively. Irrespective of the semen quality, genetic alterations were equally distributed in the SN and SN-N groups except for disomy which was significantly 3-fold higher in the SN-N as compared to the SN group.

## 4. Discussion

Our results reveal that subfertile men may be falsely classified as “normal” and a male factor contributing to genome instability may be overlooked if interpretations of semen variables only rely on the WHO reference levels. This supports the statements in the recent andrology consensus report that the WHO reference levels have reduced sensitivity to identify men with a fecundity problem [7].

Although the mean semen parameters in the SN group were well above the WHO lower reference limits [6], they were lower (and significantly for morphology) than the means observed in the F group. According to Herrera, a single parameter is not superior or inferior to the others in predicting fertility chances [44]. Coban suggested that diminished sperm quality/morphology is a potential origin for aneuploidy in early embryos [45]. The authors revealed a negative correlation between sperm morphology to chromosomal aneuploidy of the embryos in a donor oocyte program, which minimized the impact of aneuploidies arising from the female gamete focusing more on the semen parameters, particularly sperm morphology.

It has been postulated that fertile men with normal semen parameters have almost uniformly low levels of DNA breakage, whereas infertile men, especially those with compromised semen parameters, have increased proportions of nicks and breaks in the chromatin [46,47]. In our study however, in agreement with others [48,49] no differences in the levels of total SDF were observed between F, SN and SN-N groups suggesting that sperm DNA damage in semen samples may be one of the factors related to unexplained male infertility, especially in normozoospermia. Evidently, SDF may be considered an independent attribute of semen quality for all infertility patients, detecting problems not seen with semen analysis alone.

Likewise, the vital fraction did not differ between the three study groups. Although significant data are now available to suggest that higher levels of DNA damage are present in men with severe sperm defects and is an indication of a potentially negative impact on both natural and assisted conception outcomes [50] there is scarce information on this damage in the vital spermatozoa of the unfractionated sample.

Sperm DNA damage has been implicated in both male age and infertility. In a comprehensive meta-analysis by Johnson et al. [51] a decline in semen quality and an increase in SDF were associated with advancing male age. Germ cell apoptosis during spermatogenesis, which is a normal event, may be less effective in older men resulting in the release of more DNA-damaged sperm [52]. In order to avoid this excessive stress responsible for an increase in DNA damage seen with age, our study groups were limited to 40 years of age.

The recent review by Caroppo and Datillo, states that double-strand breaks are mainly associated with defective histone to protamine transition, and not with oxidative damage [53]. Moreover, sperm chromatin parameters reveal a low correlation with standard semen parameters, suggesting that these reflect completely different physiological processes during spermatogenesis [41]. Sperm nuclear maturity and chromatin stability appear to be more homogenous in a fertile population and heterogeneous in a patient population. Our F and SN groups had a mean value of 84% and 75% condensation, respectively. The SN-N group revealed a great heterogeneity with values between 8.0–93.0 %. Consequently, in vitro decondensation decreased significantly in both SN and SN-N groups as compared to the F group. Failures in condensation may induce delays in the first cell cycle with further detrimental consequences at some point for the developing embryo [24].

Incorrect chromatin compaction/hypocondensation exposes spermatozoa to DNA damage [54]. However, abnormal sperm chromatin packaging can also be manifested as a supernormal compaction which would prevent the delivery of the male genome in the oocyte [55]. Any abnormalities in the unique organization of sperm chromatin are thought to affect the proper expression and regulation of paternal genes in the early embryo [56]. In the F group, the hypo and hypercondensed ranges were low, while in the SN and SN-N groups, tremendous variations were observed.

The wide ranges in autosomal and sex chromosome aneuploidies reported among normal men may be related to the variation in the frequency of meiotic recombination [57,58] as well as abnormalities in the pairing of meiotic chromosomes [59,60]. These recombination defects may contribute to infertility in some chromosomally normal men. Whereas, infertility in some 46, XY men with subnormal semen parameters may be related to defective meiotic recombination, leading to an increased risk of aneuploidy sperm in these men [58,61,62].

The frequency of aneuploidy sperm between the SN and SN-N groups was not significantly different, denoting that hidden genetic anomalies may not be revealed just by semen analysis. Ramasamy et al. [63] studied the paternal contribution to recurrent pregnancy loss after assisted reproduction and found that 40% of the male partners with normal semen parameters also had significantly increased sperm aneuploidy levels. The inclusion of sperm aneuploidy testing might be beneficial as part of the diagnostic work-up of the infertile male.

Although sperm aneuploidy frequencies are largely consistent over time in fertile or normozöospermic males, significant differences can occur at single time points in some individuals, suggesting alterations perhaps by transient factors, lifestyle changes or environmental exposures [64]. There is a relatively linear correlation between advanced paternal age and sperm aneuploidy [65]. Increasing paternal age, together with alterations in the male endocrinal and reproductive phenotypes [66], leads to the accumulation of DNA damage over years and decreased capacity of the germ cells to repair this damage. This decline in genome integrity might lead to the production of aneuploidy sperm which translates to increased aneuploidy in embryos [67]. However, our study was controlled for age.

Although the strength of the study lies in the methodology. However, the study population could be a limitation and should be substantiated.

Approximately 20–30% of the men with normal semen parameters have an inability to achieve pregnancy [68]. Finally, our results reveal that spermatozoa from normozoospermic men, although looking perfectly normal, may still carry DNA damage, creating a problem for assisted reproductive techniques. This DNA damage may potentially modify the genetic constitution of the embryo. If the oocyte is inefficient or makes a mistake during the DNA repair process after oocyte fertilization, the potential exists to generate mutations in the embryo that will affect the pregnancy outcome and well-being of the offspring. In case of defective chromatin maturity and stability in this group, decondensation of the male pronucleus will fail after entry into the oocyte causing fertilization failures. While hidden sperm aneuploidies in men with normal semen parameters might contribute to the genetic modification of the embryo. As a consequence, we should screen subfertile men with normozoospermia for genome instability as a matter of ‘best practice’ with the aim of providing the patients with information about possible risks to their pregnancy and triggering new management strategies in reproductive medicine.

## Figures and Tables

**Figure 1 genes-14-00239-f001:**
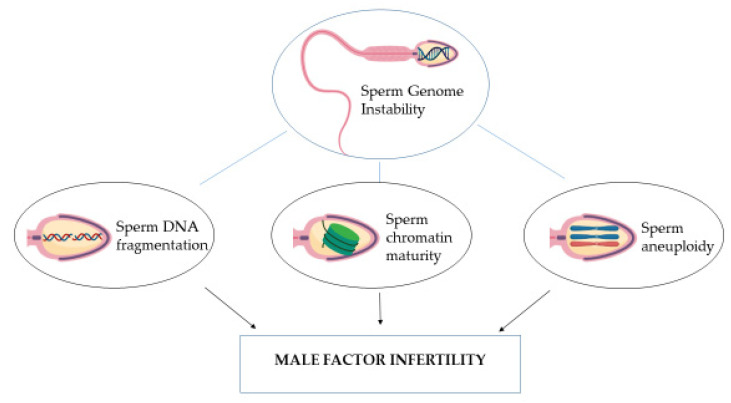
Schematic representation of genome instability assessment in male factor infertility. Figure created with BioRender.com.

**Figure 2 genes-14-00239-f002:**
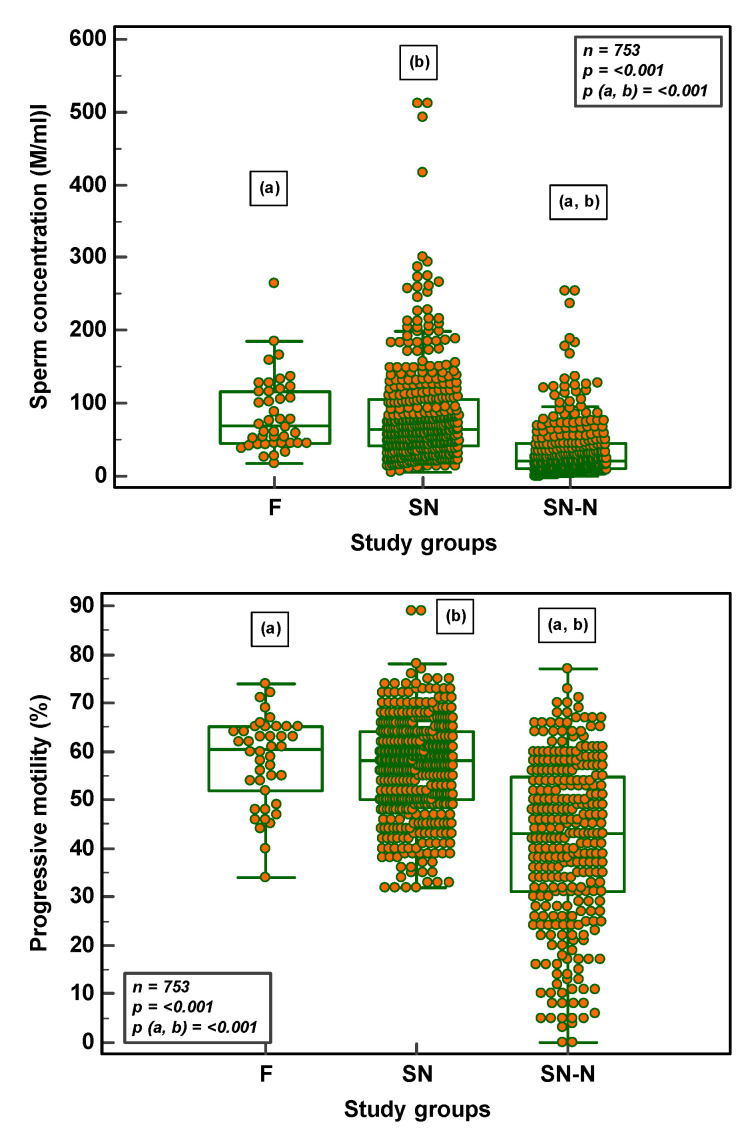
Semen parameters in the different study groups (F = fertile population; SN = subfertile population with normozoospermia; SN-N = subfertile non-normozoospermic population).

**Figure 3 genes-14-00239-f003:**
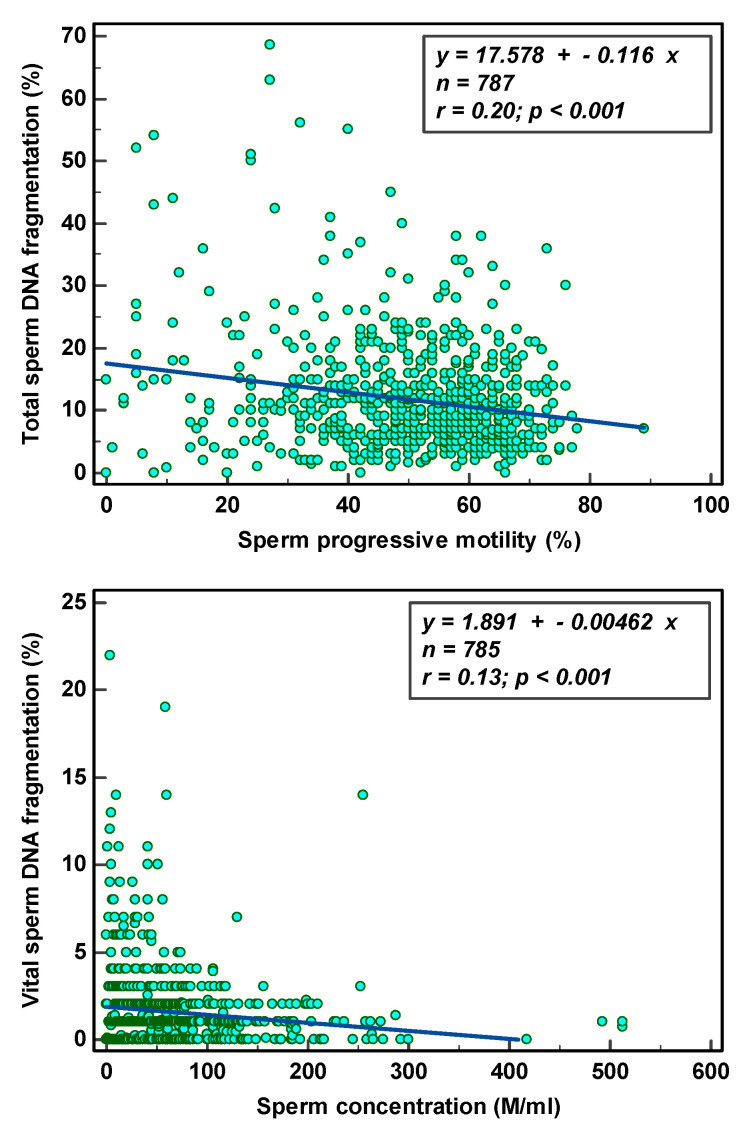
Scatter diagram with regression line between SDF parameters and semen parameters.

**Figure 4 genes-14-00239-f004:**
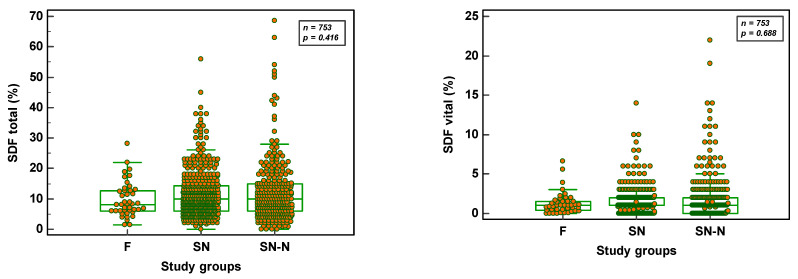
Total and vital SDF parameters in the different study groups (F = fertile population; SN = subfertile population with normozoospermia; SN-N = subfertile non-normozoospermic population).

**Figure 5 genes-14-00239-f005:**
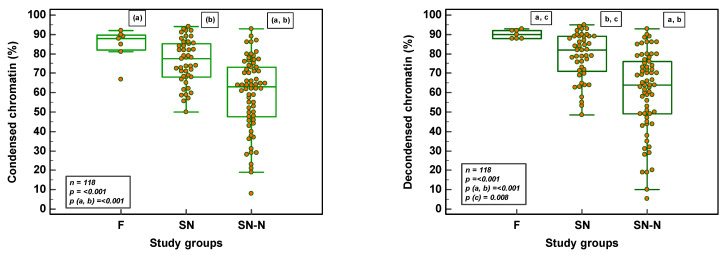
Chromatin parameters in the different study groups (F = fertile population; SN = subfertile population with normozoospermia; SN-N = subfertile non-normozoospermic population).

**Figure 6 genes-14-00239-f006:**
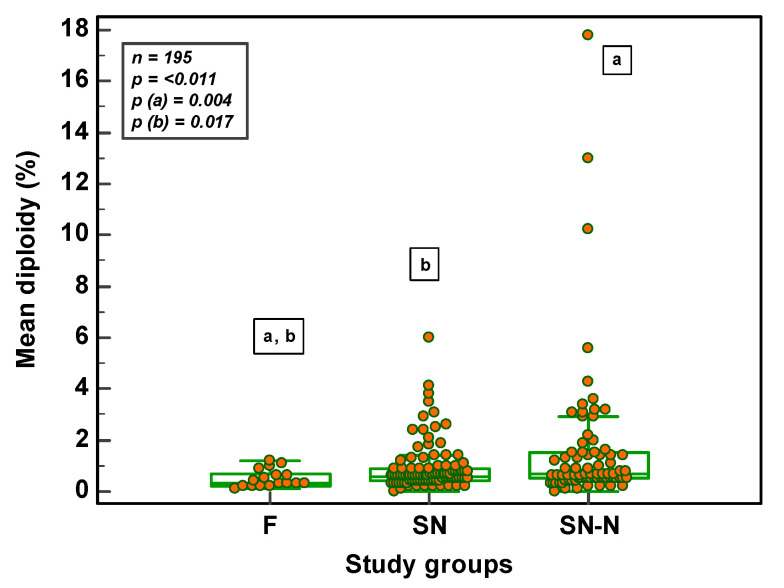
Frequency of mean sperm diploidy in the different study groups (F = fertile population; SN = subfertile population with normozoospermia; SN-N = subfertile non-normozoospermic population).

**Figure 7 genes-14-00239-f007:**
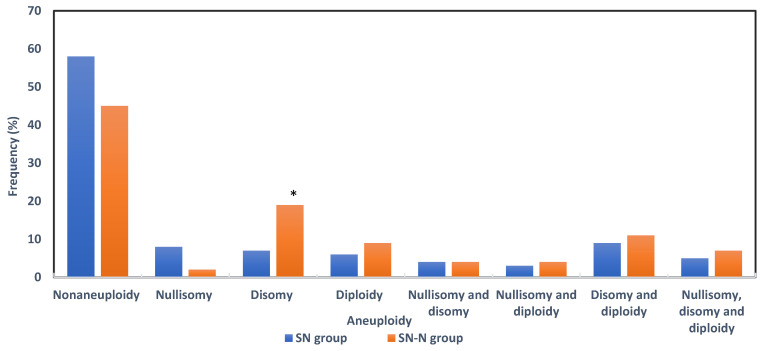
Frequency of aneuploidies after FISH analysis (SN = subfertile population with normozoospermia; SN-N = subfertile non-normozoospermic population). * Fischer’s exact test *p* = 0.0108.

**Table 1 genes-14-00239-t001:** Sperm aneuploidy in the fertile and subfertile groups (F = fertile population; SN = subfertile population with normozoospermia; SN-N = subfertile non-normozoospermic population).

Sperm Aneuploidy	Study Groups			*p* Value *
	F (*n* = 17)	SN (*n* = 100)	SN-N (*n* = 78)	
Chromosome 13				
nullisomy (%)	0.12 ± 0.16	0.17 ± 0.27	0.13 ± 0.16	0.632
disomy (%)	0.15± 0.11	0.15 ± 0.28	0.18 ± 0.29	0.188
Chromosome 18				
nullisomy (%)	0.05 ± 0.09	0.21 ± 0.50	0.13 ± 0.22	0.262
disomy (%)	0.11 ± 0.17	0.22 ± 0.58	0.19 ± 0.25	0.237
Chromosome 21				
nullisomy (%)	0.08 ± 0.12	0.14 ± 0.15	0.12 ± 0.17	0.173
disomy (%)	0.09 ± 0.09	0.15 ± 0.40	0.19 ± 0.30	0.141
Chromosome X/Y				
nullisomy (%)	0.28 ± 0.31	0.26 ± 0.39	0.31 ± 0.54	0.329
disomy XX (%)	0.15 ± 0.22	0.11 ± 0.15	0.10 ± 0.17	0.605
disomy XY (%)	0.11 ± 0.13	0.24 ± 0.89	0.24 ± 0.34	0.376
disomy YY (%)	0.06 ± 0.10	0.07 ± 0.16	0.07 ± 0.13	0.648
Autosomal aneuploidy (%)	0.60 ± 0.35	1.04 ± 1.24	0.95 ± 0.84	0.347
Sex aneuploidy (%)	0.61 ± 0.42	0.68 ± 1.14	0.72 ± 0.86	0.605
Diploidy (%)	0.49 ± 0.35	0.93 ±01.95	1.59 ± 2.73	0.011

Data are presented as mean ± SD; * Kruskal–Wallis test.

## Data Availability

The data presented in this study are available from the corresponding author at a reasonable request.

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
