# Peer review of "Semen Thresholds of Normality Established by the WHO Do Not Reveal Genome Instability—A Potential Occult Male Factor"

_genes, 2023, doi:10.3390/genes14020239_

Round 1
Reviewer 1 Report
The authors tried to suggest the importance of genome instability in male infertility, however, the data was insufficient. Only chromatin maturity and stability was significantly different in the three groups, while the SDF parameters and sperm aneuploidy showed no difference. In addition, some parts are also confusing. For example, the causality and logic of Figure 1 should be reconsidered. The analysis of semen parameters in fertile, subfertile normozoospermic and subfertile non-normozoo-spermic males was meaningless, because grouping is based on semen parameters. Why only chose 13, 18, 21 and X/Y? What is the threshold of chromatin maturity and stability for male infertility?
Author Response
Dear Reviewer,
Manuscript ID: genes-2138876
Title: Semen thresholds of normality established by the WHO do not reveal genome instability – a potential occult male factor
First of all we would like to express our deep sense of gratitude to the reviewers for the time and energy spent in reading and reviewing our study. We have thoroughly revised and modified the manuscript where necessary and we hope that the revised edition will meet the expectations of the reviewers.
All revisions made to the manuscript have been done so using the “Track Changes” function. We have also carried out a spelling and grammatical check for the English language and the style modified where required. The relevancy and the correctness of the references used have also been controlled.
Response to reviewer #1
Must be improved: Research design improved on Page 2; lines 95 - 98
Methods adequately described improved on Page 4; lines 171 - 183
Conclusions modified on Pages 12 - 13; lines 445 – 460
Comment: The authors tried to suggest the importance of genome instability in male infertility, however, the data was insufficient. Only chromatin maturity and stability was significantly different in the three groups, while the SDF parameters and sperm aneuploidy showed no difference
Answer: We understand the reviewers concern over the statistical significance revealed only in chromatin maturity and stability. What we do have to emphasize is that lack of statistical significance in SDF and sperm aneuploidy should not blind us to the dangers the broad ranges present in the SN group might lead to. As to the insufficient data, we have taken this up as a limitation. The following text is included:
Although the strength of the study lies in the methodology. However, the study population could be a limitation and should be substantiated.
Approximately 20 – 30% of the men with normal semen parameters have inability to achieve pregnancy [68]. Finally our results reveal that spermatozoa from normozoospermic men, although looking perfectly normal, may still carry DNA damage, creating a problem for assisted reproductive techniques. This DNA damage may potentially modify the genetic constitution of the embryo. If the oocyte is inefficient or makes a mistake during the DNA repair process after oocyte fertilization, the potential exists to generate mutations in the embryo that will affect the pregnancy outcome and well-being of the offspring. In case of defective chromatin maturity and stability in this group, decondensation of the male pronucleus will fail after entry into the oocyte causing fertilization failures. While hidden sperm aneuploidies in men with normal semen parameters might contribute to the genetic modification of the embryo. As a consequence, we should screen subfertile men with normozoospermia for genome instability as a matter of ‘best practice’ with the aim of providing the patients with information about possible risks to their pregnancy and to trigger new management strategies in reproductive medicine.
Pages 12-13 ; lines 445 – 460: modified
Comment: In addition, some parts are also confusing. For example, the causality and logic of Figure 1 should be reconsidered.
Answer: The authors are grateful for this suggestion and have edited and modified the text as suggested.
Genome instability in sperm can be accessed via SDF, chromatin maturity and aneuploidy which might potentially lead to a male factor infertility as schematically presented in Figure 1.
Page 5; lines 221 – 224: modified
Comment: The analysis of semen parameters in fertile, subfertile normozoospermic and subfertile non-normozoo-spermic males was meaningless, because grouping is based on semen parameters.
Answer: The reviewer is right in this observation. But, semen analysis remains a cornerstone in our diagnostic andrology clinic. So we took the privilege to present the data.
Comment: Why only chose 13, 18, 21 and X/Y?
Answer: We selected 13, 18, 21 and X/Y chromosomes as numerical alterations in these chromosomes are compatible with life but cause several syndromes.
Comment: What is the threshold of chromatin maturity and stability for male infertility?
Answer: The thresholds obtained can vary with the methodologies applied. Using our method we have published the associated criterion for each chromatin parameter (Punjabi et al., 2019):
Condensed chromatin > 77%; hypocondensed ≤ 10% and hypercondensed ≤ 4%
We thank you in advance for considering our work and look forward to hearing from you.
Yours sincerely,
Usha Punjabi
Reviewer 2 Report
Review Manuscript ID: genes-2138876, entitled “Semen thresholds of normality established by the WHO do not reveal genome instability – a potential occult male factor”
In the submitted manuscript, the study was undertaken to asses sperm genome stability in subfertile males occurring in spite of a normal semen analyse.
I have few points, which in my opinion should be explained:
L115-127 how many people were assessed in each group?
L167 ,semen samples were aliquoted into two fractions’ I do not think this is a very precise term. It would be necessary to specify more precisely what semen volume and sperm concentration are concerned?
L173 as above
Figure 2. 753 samples were evaluated: but it is not known how many were in each study group.
There is no clear summary resulting from the research conducted. What conclusions can be drawn from these studies?
Author Response
Dear Editor,
Manuscript ID: genes-2138876
Title: Semen thresholds of normality established by the WHO do not reveal genome instability – a potential occult male factor
First of all we would like to express our deep sense of gratitude to the reviewers for the time and energy spent in reading and reviewing our study. We have thoroughly revised and modified the manuscript where necessary and we hope that the revised edition will meet the expectations of the reviewers.
All revisions made to the manuscript have been done so using the “Track Changes” function. We have also carried out a spelling and grammatical check for the English language and the style modified where required. The relevancy and the correctness of the references used have also been controlled.
Response to reviewer #2
Must be improved: Research design improved on Page 2; lines 95 - 98
Methods adequately described improved on Page 4; lines 171 - 183
Conclusions modified on Pages 12 - 13; lines 445 – 460
Comment: L115-127 how many people were assessed in each group?
Answer: Our sincere excuses for the confusion created. The number of samples analysed in each group is now added as follows::
In the semen samples analysed (F = 42; SN = 416; SN-N = 295), there was a significant difference (p <0.001) between the three study groups as far as sperm concentration, motility and morphology was concerned.
Page 6; line 266: modified
Comment: L167 ,semen samples were aliquoted into two fractions’ I do not think this is a very precise term. It would be necessary to specify more precisely what semen volume and sperm concentration are concerned?
Answer: The authors appreciate the reviewers suggestion and have made the following modifications:
In brief, an aliquot of the semen sample is added to two test tubes containing 1ml Tris buffer to achieve a concentration of ± 5 x 106 spermatozoa per ml. After washing and centrifugation, one test tube was treated with the DNA-intercalating dye propidium iodide (Sigma-Aldrich, Belgium, PI, 50µg/ml) followed by flow cytometric evaluation of the PI fluorescence intensity on a cell per cell basis.
Page 4; lines 171 - 174: modified
Comment: L173 as above
Answer: The reviewers suggestion has been followed here:
The second test tube was treated with 1% sodium dodecyl sulphate (SDS, Sigma-Aldrich, Belgium) plus 6 mmol/l ethylene diamine tetra acetic acid (EDTA, Sigma-Aldrich, Belgium) decondensing solution in borate buffer (Sigma-Aldrich, Belgium) for 5 mins. Following which the sample was washed and centrifuged before using PI.
Page 4; lines 179 - 183: modified
Comment: Figure 2. 753 samples were evaluated: but it is not known how many were in each study group.
Answer: Our sincere excuses for the confusion created. The number of samples analysed in each group is now added as follows::
In the semen samples analysed (F = 42; SN = 416; SN-N = 295), there was a significant difference (p <0.001) between the three study groups as far as sperm concentration, motility and morphology was concerned.
Page 6; line 266: modified
Comment: There is no clear summary resulting from the research conducted. What conclusions can be drawn from these studies?
Answer: We appreciate the reviewers attempt in making us alert to provide a conclusion. The following has been added:
Although the strength of the study lies in the methodology. However, the study population could be a limitation and should be substantiated.
Approximately 20 – 30% of the men with normal semen parameters have inability to achieve pregnancy [68]. Finally our results reveal that spermatozoa from normozoospermic men, although looking perfectly normal, may still carry DNA damage, creating a problem for assisted reproductive techniques, such as ICSI. This DNA damage may potentially modify the genetic constitution of the embryo. If the oocyte is inefficient or makes a mistake during the DNA repair process after oocyte fertilization, the potential exists to generate mutations in the embryo that will affect the pregnancy outcome and well-being of the offspring. In case of defective chromatin maturity and stability in this group, decondensation of the male pronucleus will fail after entry into the oocyte causing fertilization failures. While hidden sperm aneuploidies in men with normal semen parameters might contribute to the genetic modification of the embryo. As a consequence, we should screen subfertile men with normozoospermia for genome instability as a matter of ‘best practice’ with the aim of providing the patients with information about possible risks to their pregnancy and to trigger new management strategies in reproductive medicine.
Pages 12 - 13; lines 445 – 460
We thank you in advance for considering our work and look forward to hearing from you.
Yours sincerely,
Usha Punjabi
Round 2
Reviewer 1 Report
I think the authors have made a great improvement to the article. I have no suggestion now.
Reviewer 2 Report
The authors have corrected the article according to my comments. I believe that the work can be published in Genes.